# The effect of adjuvants and delivery systems on Th1, Th2, Th17 and Treg cytokine responses in mice immunized with *Mycobacterium tuberculosis*-specific proteins

**Hussain A. Safar** [1], **Abu Salim Mustafa** [1]*, **Hanady A. Amoudy** [1], **Ahmed El-Hashim** [2]

**1** Department of Microbiology, Faculty of Medicine, Kuwait University, Kuwait City, Kuwait, **2** Department of Pharmacology & Therapeutics, Faculty of Pharmacy, Kuwait City, Kuwait

* abusalim@hsc.edu.kw

**Data Availability Statement:** All relevant data are within the manuscript and its Supporting Information files.

## Abstract

Tuberculosis (TB) is a major health problem of global concern. The control of this disease requires appropriate preventive measures, including vaccines. In TB, T helper (Th)1 cytokines provide protection whereas Th2 and T regulatory (Treg) cytokines contribute to the pathogenesis and Th17 cytokines play a role in both protection and pathogenesis. Previous studies with *Mycobacterium tuberculosis*-specific proteins have identified seven low molecular weight proteins, PE35, ESXA, ESXB, Rv2346c, Rv2347c, Rv3619c, and Rv3620c, as immunodominant antigens inducing Th1-cell responses in humans following natural infection with *M. tuberculosis*. The aim of this study was to characterize the cytokine responses induced in mice immunized with these proteins, using various adjuvants and delivery systems, i.e. chemical adjuvants (Alum and IFA), non-pathogenic mycobacteria (*M. smegmatis* and *M. vaccae*) and a DNA vaccine plasmid (pUMVC6). The immune responses were monitored by quantifying the marker cytokines secreted by Th1 (IFN-γ), Th2 (IL-5), Treg (IL-10), and Th17 (IL-17A) cells. DNA corresponding to *pe35*, *esxa*, *esxb*, *rv2346c*, *rv2347c*, *rv3619c*, and *rv3620c* genes were cloned into the expression vectors pGES-TH-1, pDE22 and pUMVC6 for expression in *Escherichia coli*, mycobacteria and eukaryotic cells, respectively. Mice were immunized with the recombinants using different adjuvants and delivery systems, and spleen cells were stimulated in vitro with peptides of immunizing proteins to investigate antigen-specific secretion of Th1 (IFN-γ), Th2 (IL-5), Treg (IL-10), and Th17 (IL-17A) cytokines. The results showed that spleen cells, from mice immunized with all antigens, secreted the protective Th1 cytokine IFN-γ, except ESXB, with one or more adjuvants and delivery systems. However, only Rv3619c consistently induced Th1-biased responses, without the secretion of significant concentrations of Th2, Th17 and Treg cytokines, with all adjuvants and delivery systems. Rv3619c also induced antigen-specific IgG antibodies in immunized mice.

**Funding:** ASM HAA AE HAS: A graduate research grant from the College of Graduate Studies, Kuwait University (kuniv.edu/COGS/index.htm) ASM HAA AE HAS: grant numbers YM06/15 and SRU02/13 from the Research Sector, Kuwait University (www.ovpr.kuniv.edu/) ASM HAS: grant number CB17-63MM-03 from Kuwait Foundation for the Advancement of Sciences (kfas.org.kw/) The funders had no role in study design, data collection and analysis, decision to publish, or preparation of the manuscript.

**Competing interests:** The authors have declared that no competing interests exist.

## Introduction

Tuberculosis (TB) is the tenth leading cause of global deaths worldwide and the leading cause of death due to a single infectious disease thus ranking it above HIV/AIDS [1]. According to the most recent report from the WHO, about 10 million people suffered from TB complications and approximately 1.3 million people died of TB in 2017 [1]. Approximately a quarter of the world population is latently infected with *Mycobacterium tuberculosis* [2]. However, not all individuals infected with *M. tuberculosis* present with the symptoms of the disease and in fact the majority of infected people are clinically asymptomatic. Nonetheless, they are at a significant risk of developing active TB when their immune system becomes weakened or compromised following reactivation of latent *M. tuberculosis* [3, 4]. The End TB strategy of the WHO aims to reduce TB morbidity by 90% and TB mortality by 95% by 2035 [1]. This objective can only be achieved when robust diagnostic technologies, new therapies with a short-course nature, and effective vaccines are factorized and delivered [5].

The only available vaccine for human use is the attenuated and live *M. bovis* Bacillus Calmette Guerin (BCG), which was introduced to humans in 1920s. Although BCG has been used extensively in large parts of the world, it has failed to provide consistent protective efficacy in humans, particularly in the developing world and against adult pulmonary disease, the most common manifestation of TB [1, 6]. In addition, BCG vaccination has other drawbacks including disseminated BCGosis in immunocompromised individuals, e.g. HIV-infected [7]. Furthermore, the efficacy of BCG is reduced in individuals pre-sensitized with environmental mycobacteria due to the presence of crossreactive antigens [6, 8]. Therefore, the attention has been focused to develop new vaccines that can complement or replace BCG [9].

Among the new vaccine options are subunit vaccines based on *M. tuberculosis*-specific antigens, which can improve BCG-induced protection [10]. These subunit vaccines, known as booster vaccines, are administered to individuals who are pre-vaccinated with BCG or pre-sensitized with environmental mycobacteria, to enhance the induced protection [10]. Subunit vaccines can also be used for therapeutic applications by aiding the immune system to eliminate *M. tuberculosis* [11]. The sequencing of *M. tuberculosis* genome and advances in comparative genomics and bioinformatics have helped to identify species-specific genomic regions and encoded proteins in different mycobacterial species [6, 12, 13]. By comparing mycobacterial genomes, it was shown that 16 regions of differences (RD)1 to RD16 existed among *M. tuberculosis*, *M. bovis* and BCG [14, 15]. Among these RDs, 11 *M. tuberculosis*-specific genomic regions (RD1, RD4-7, RD9-RD13, and RD15) were absent/deleted in all BCG strains [16]. These regions were predicted to encode 89 proteins [16]. When tested with human peripheral blood mononuclear cells, seven low molecular weight proteins encoded by RD1 (PE35, ESXA and ESXB), RD7 (Rv2346c and Rv2347c) and RD9 (Rv3619c and Rv3620c) induced the best Th1 cell responses (IL-2 and IFN-γ) [17, 18, 19].

The identification of major *M. tuberculosis*-specific antigens has paved the way for studying their role in inducing protective immunity against TB [20]. However, the delivery of these antigens is a key issue that may have influence on the quality of immune responses induced, i.e. protective (Th1) [21, 22], pathologic (Th2 and Treg) [23, 24], or both protective and pathologic (Th17) [25]. To induce the desired immune responses against protein antigens, adjuvants and delivery systems are often required to deliver the antigens [26]. Choosing an appropriate adjuvant and delivery platform is critical for the induction of Th1-biased responses [27], and thus important for the construction of an effective subunit vaccine. The aim of this work was to evaluate chemical adjuvants (Aluminum hydroxide [Alum] and (Freund's Incomplete Adjuvant [IFA]), live non-pathogenic mycobacteria (*M. smegmatis* and *M. vaccae*) and a DNA vaccine vector (pUMVC6) as adjuvants and delivery systems to study the induction of Th1,

Th2, Th17 and Treg cytokines following immunization with seven low molecular weight *M. tuberculosis*-specific antigens, i.e. PE35, ESXA, ESXB, Rv2346c, Rv2347c, Rv3619c and Rv2360c, in mice.

# Materials and methods

## Plasmid vectors and bacterial strains

The plasmids pGEM-T Easy (Promega corporation Madison, WI, USA), pDE22 [24] and DNA vaccine vector pUMVC6 (Aldevron, Fargo, North Dakota, USA) were propagated in *Escherichia coli* strain TOP10 (ATCC, Manassas, VA, USA), and the plasmid pGESTH-1) [28] was propagated in *E. coli* strain BL-21 (Novagen, Madison, WI, USA), as described previously [29, 30]. The shuttle vector pDE22 was used for the expression of *pe35*, *esxa*, *esxb*, *rv2346c*, *rv2347c*, *rv3619c* and *rv3620c* genes in *M. smegmatis* and *M. vaccae*, as described previously [24, 31]. Genomic DNA isolated from *M. tuberculosis* H37Rv (obtained from the American Type Culture Collection, (Rockville, MD, USA) served as the source for the amplification and subsequent cloning of the genes, as previously described [28, 30]. All DNA manipulations, restriction enzyme digestions and bacterial cell transformations with the plasmids were performed according to previously described procedures [28–32].

## PCR primers

The primers for amplifications of target genes from the genomic DNA of *M. tuberculosis* by polymerase chain reaction (PCR) were designed on the basis of nucleotide sequences of these genes in *M. tuberculosis* genome (Tuberculist–Genolist Institute Pasteur, http://genolist. pasteur.fr/TubercuList/). The nucleotide sequences of each forward (F) and reverse (R) primer are given in Tables 1 and 2. All primers contained additional sequences at the 5' end for digestion with appropriate restriction enzymes to clone efficiently the PCR-amplified DNA in the various vectors, i.e. pGEM-T Easy, pGES-TH-1, pDE22 and pUMVC6, as previously described [23, 24, 28–32]. The primers were synthesized commercially (ThrmoFisher Scientific, Ulm, Germany).

## Cloning of genes in various vectors

DNA corresponding to *pe35*, *esxa*, *esxb*, *rv2346c*, *rv2347c*, *rv3619c* and *rv3620c* genes were amplified by PCR using genomic DNA isolated from *M. tuberculosis* and gene-specific primers, as previously described [29, 31, 32]. The amplified DNA were ligated to the cloning vector pGEM-T Easy and propagated in *E. coli* TOP10. The identities of genes cloned in pGEM-T

**Table 1. Nucleotide sequences of forward and reverse primers used for the amplification of genes from the genomic DNA of *M. tuberculosis* and cloning of the amplified products in pGEM-T Easy, pGES-TH-1 and pDE22 vectors.**

| Gene | Nucleotide sequences of forward primers | Nucleotide sequences of reverse primers |
|------|------------------------------------------|------------------------------------------|
| *pe35* | 5'-aatcggatccatggaaaaaatgtcacatgatccg-3 | 5' acgaagctttcggcgaagacgccggcggcgccgt 3' |
| *esxa* | 5' aatcggatccatgacagagcagcagtggaatttc 3' | 5' acgaagctttgcgaacatcccagtgacgtt 3' |
| *esxb* | 5' aatcggatccatggcagagatgaagaccgatgcc 3' | 5' acgaagcttgaagcccatttgcgaggacag 3' |
| *rv2346c* | 5' aatcggatccatgaccatcaactatcagttcggt 3' | 5' acgaagcttggcccagctggagccgacggcgct 3' |
| *rv2347c* | 5' aatcggatccatggcaacacgtttttatgacggat 3' | 5' acgaagcttgctgctgaggatctgctgctgggaggc 3' |
| *rv3619c* | 5' aatcggatccatgaccatcaactatcaattcggg 3' | 5' acgaagcttggcccagctggagccgacggcgct 3' |
| *rv3620c* | 5' aatcggatccatgacctcgcgtttttatgacggat 3' | 5' acgaagcttgctgctgaggatctgctgctgggaggc 3' |

The restriction sites for *Bam*H I and *Hind* III are underlined in the forward and reverse primers, respectively.

**Table 2. Nucleotide sequences of forward and reverse primers used for the amplification of genes from the genomic DNA of *M. tuberculosis* and cloning of the amplified products in pUMVC6 vector.**

| Gene | Nucleotide sequences of forward primers | Nucleotide sequences of reverse primers |
|---|---|---|
| *pe35* | 5'-aatcggatccatggaaaaaatgtcacatgatccg-3 | 5- acgggatccgaagcccatttgcgaggacag -3' |
| *esxa* | 5' aatcggatccatgacagagcagcagtggaatttc 3' | 5' acgggatcctgcgaacatcccagtgacgtt 3' |
| *esxb* | 5' aatcggatccatggcagagatgaagaccgatgcc 3' | 5' acgggatccgaagcccatttgcgaggacag 3' |
| *rv2346c* | 5' aatcggatccatgaccatcaactatcagttcggt 3' | 5' acgggatccggcccagctggagccgacggcgct 3' |
| *rv2347c* | 5' aatcggatccatggcaacacgttttatgacggat 3' | 5' acgggatccgctgctgaggatctgctgctgggaggc 3' |
| *rv3619* | 5' aatcggatccatgaccatcaactatcaattcggg 3' | 5' acgggatccggcccagctggagccgacggcgct 3' |
| *rv3620* | 5' aatcggatccatgacctcgcgttttatgacggat 3' | 5' acgggatccgctgctgaggatctgctgctgggaggc 3' |

The restriction sites for *Bam*H I are underlined in the forward and reverse primers.

Easy were determined by restriction digestion and DNA sequencing according to standard procedures [31]. The DNA fragments corresponding to the amplified genes were restriction digested from the recombinant pGEM-T Easy and subsequently cloned into the expression vector pGES-TH-1, shuttle vector pDE22 and DNA vaccine vector pUMVC6, as described previously [28, 29, 31].

## Recombinant proteins

The expression vector pGES-TH-1 was used for high level expression of PE35, ESXA, ESXB, Rv2346c, Rv2347c, Rv3619c and Rv3620c fusion proteins in *E. coli*, as described previously [28, 29]. The expression of fusion proteins was determined by western blotting using anti-GST antibodies [28]. The recombinant proteins were purified to homogeneity using two affinity columns, i.e. glutathione Sepharose column and Ni-NTA agarose column [28].

## Mitogen and synthetic peptides

The mitogen Concanavalin A (ConA) was purchased from Sigma Chemicals, St. Louis, MO, USA. The peptides (25-mer overlapping with neighboring peptides by 10 residues) covering the sequences of PE35, ESXA, ESXB, Rv2346c, Rv2347c, Rv3619c and Rv3620c proteins were synthesized by solid phase peptide synthesis using fluorenylmethoxycarbonyl chemistry [31]. The peptides were dissolved in sterile phosphate-buffered saline (pH 7.0) and frozen at -20°C in aliquots, as described previously [31].

## Recombinant mycobacteria

The recombinant plasmids pDE22/*esxa*, pDE22/*esxb*, pDE22/*pe35*, pDE22/*rv2346c*, pDE22/*rv2347c*, pDE22/*rv3619c* and pDE22/*rv3620c* were electroporated into *M. smegmatis* and *M. vaccae* and the expression of genes in recombinant (r)*M. smegmatis* and r*M. vaccae* was determined by reverse-transcriptase (RT)-PCR, as described previously [24].

## Experimental animals

Six to eight weeks old female pathogen-free BALB/c mice were used in this study. All experiments in mice were performed in accordance with the principles of NC3Rs' ARRIVE guidelines for reporting humane animal research, the BJP Guidelines and in accordance with the EU Directive 2010/63/EU for animal experiments and the National Institutes of Health guide for the care and use of laboratory animals (NIH Publications No. 8023, revised 1978). The experimental protocols were approved by the "Health Science Center Animal Welfare

Committee" and complied with regulations for the animal care and ethical use of Laboratory Animals in the Health Sciences Center, Kuwait University. The mice were divided into 35 groups (n = 5 mice per group), and one group of mice was immunized with each *M. tuberculosis*-specific antigen along with a given delivery system.

## Immunization of mice

Fourteen groups of mice were immunized intra-peritoneally with 2 μg of purified recombinant proteins emulsified with IFA (Sigma-Aldrich, St. Louis, MO, USA) or adsorbed onto Alum (ThermoFisher Scientific Inc., Waltham, MA, USA) and boosted twice with the proteins at two-week intervals, as described previously [33]. Furthermore, 14 groups of mice were immunized and boosted three times (at two-week intervals) intra-peritoneally with $5x10^7$ colony forming units (CFU) of r*M. smegmatis* or r*M. vaccae* expressing the cloned genes, as described previously [24, 31]. In addition, seven groups of mice were immunized and boosted twice (at three-week interval) intramuscularly with 100 μg rDNA vaccine constructs, as described previously [29, 34]. Two weeks (in case of chemical adjuvants [IFA and Alum] and recombinant mycobacteria groups) and three weeks (in case of rDNA vaccine groups) after the last booster, mice were euthanized, and spleen cells were collected aseptically according to standard procedures [29, 32].

## Spleen cell cultures for cytokine assays

Mitogen and peptide-induced secretions of cytokines from mouse spleen cells were determined according to standard procedures [31, 32]. In brief, spleen cells were seeded into 96-well tissue culture plates (Nunc, Roskilde, Denmark) and stimulated in triplicates with ConA (experimental positive control) and the pool of peptides (PPs) covering the sequence of individual immunizing proteins. The stimulants ConA and PPs were used at optimal concentrations, i.e. 4 μg/ml and 25 μg/ml, respectively [29]. The negative control wells didn't receive any of these stimulants to test for antigen-nonspecific secretion of cytokines. The culture plates were incubated at 37˚C in a humidified atmosphere of 5% $CO_2$ and 95% air. The culture supernatants were collected for cytokine assays, after appropriate time intervals, as described previously [24]. The culture supernatants of spleen cells from each group of mice (n = 5) stimulated with the mitogen or PPs were pooled and tested in duplicates for concentrations of Th1 cytokine IFN-γ, Th2 cytokine IL-5, Treg cytokine IL-10, and Th17 cytokine IL-17A using enzyme-linked immunosorbent assay (ELISA) kits (ThermoFisher Scientific Inc, Waltham, MA, USA), as specified by the manufacturer. The average cytokine concentrations from the duplicates were calculated. The minimum detectable concentrations of IFN-γ, IL-5, IL-10, and IL-17A by using the kits were 1, 1, 1, and 0.5 pg/ml, respectively.

Th1, Th2, Treg, Th17 bias or no bias were calculated from the ratios of IFN-γ:IL-5, IFN-γ:IL-10 and IFN-γ:IL-17A concentrations [17]. The ratios exceeding 2 were considered as Th1 bias, 1 to 2 no bias, <1 for IFN-γ:IL-5 as Th2 bias, <1 for IFN-γ:IL-10 as Treg bias and <1 for IFN-γ:IL-17A as Th17 bias [17]. A given Th bias was considered predominant with >50% antigen and adjuvant/delivery system combinations inducing it.

## Data analysis

All results were analyzed using GraphPad Prism (GraphPad Software, San Deigo, CA, USA). The concentrations of cytokines in response to various stimulants were considered significant with quantities >100 pg/ml and the ratio of Experimental/ Negative control (E/C)> 2 [31, 35]. The objective of comparison of cytokine data in all figures was to determine which

 

immunization, followed by stimulation of spleen cells in vitro, resulted in secretion of significant concentrations of the individual cytokines.

## Results

### Spontaneous and mitogen-induced secretion of Th1 (IFN-γ), Th2 (IL-5), Th17 (IL-17A) and Treg (IL-10) cytokines by spleen cells

Spontaneous secretion of various cytokines by spleen cells in the absence of exogenously added stimulants varied from 1 to 1657 pg/ml (Figs 1–5). On the other hand, significant concentrations of all cytokines were secreted from spleen cells stimulated with ConA (S1 Fig). These results suggested that the culture conditions were appropriate.

### Characterization of the immune responses (in terms of Th1, Th2, Th17 and Treg cytokines) to *M. tuberculosis*-specific antigens in mice immunized with the various delivery systems

The quantification of secreted cytokines from spleen cells of mice in response to PPs showed that spleen cells of mice immunized with Rv2347c/Alum and Rv3619c/Alum (Fig 1); PE35/IFA, Rv2346c/IFA, Rv2347c/IFA, Rv3619c/IFA (Fig 2) secreted IFN-γ. Interestingly, Th2 (IL-5), Treg (IL-10), and Th17 (IL-17A) cytokines were not detected in these groups, except for IL-5 which was detected in mice immunized with ESXA/Alum (Fig 1). As for recombinant mycobacteria, spleen cells secreted IFN-γ from mice immunized with r*M. smegmatis*/*rv2346c*, r*M. smegmatis*/ *rv2347c*, r*M. smegmatis*/*rv3619c* (Fig 3); and r*M. vaccae*/*pe35*, r*M. vaccae*/*esxa*, r*M. vaccae*/ *rv2347c*, r*M. vaccae*/*rv3619c* and r*M. vaccae*/ *rv3620c* (Fig 4). Furthermore, IL-5 and IL-10 were not secreted from spleen cells of these mice and IL-17A was secreted from spleen cells of mice immunized with r*M. vaccae*/ *rv2346c* only. In addition, the spleen cells of mice immunized with recombinant DNA vaccine constructs of PE35, ESXA, Rv3619c and

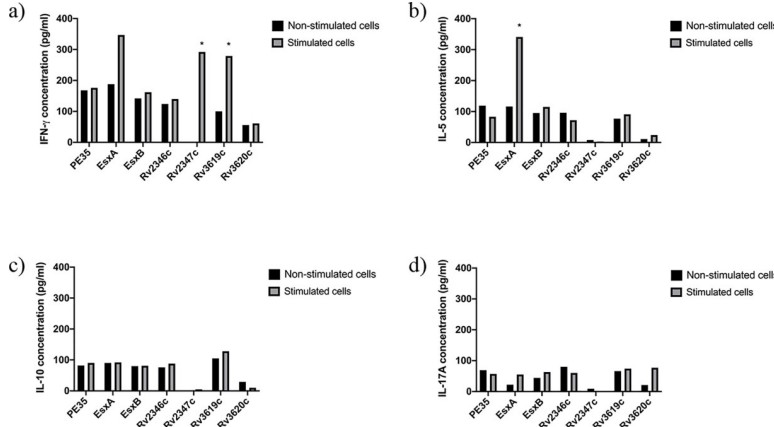

**Fig 1. Concentrations (pg/ml) of IFN-γ, IL-5, IL-10 and IL-17A in pools of culture supernatants of spleen cells obtained from mice (n = 5 per group) immunized with purified recombinant proteins adsorbed onto Alum.** Spleen cells obtained from immunized mice were cultured in triplicates in the absence of any stimulant (negative control) and in the presence of stimulants (experimental), i.e. pools of peptides (PPs) covering the sequences of individual immunizing proteins. The supernatants were collected on day 6. The culture supernatants from triplicates of negative controls and each stimulant were pooled separately and tested for secreted a) IFN-γ, b) IL-5, c) IL-10, and d) IL-17A cytokines in duplicate wells of 96-well plates by ELISA. The average values of cytokine concentrations from duplicates are presented in the figure. The concentrations of cytokines in response to various stimulants were considered significant with quantities >100 pg/ml and the ratio of Experimental/ Negative control (E/C)> 2 [31, 35]. Such values are marked with an asterisk (*).

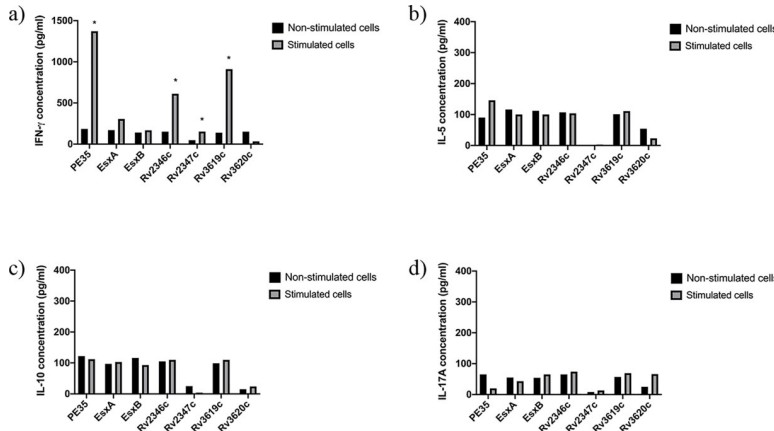

**Fig 2. Concentrations (pg/ml) of IFN-γ, IL-5, IL-10 and IL-17A in pools of culture supernatants of spleen cells obtained from mice (n = 5 per group) immunized with purified recombinant proteins emulsified with IFA.** Spleen cells obtained from immunized mice were cultured in triplicates in the absence of any stimulant (negative control) and in the presence of stimulants (experimental), i.e. pools of peptides (PPs) covering the sequences of individual immunizing proteins. The supernatants were collected on day 6. The culture supernatants from triplicates of negative controls and each stimulant were pooled separately and tested for secreted a) IFN-γ, b) IL-5, c) IL-10, and d) IL-17A cytokines in duplicate wells of 96-well plates by ELISA. The average values of cytokine concentrations from duplicates are presented in the figure. The concentrations of cytokines in response to various stimulants were considered significant with quantities >100 pg/ml and the ratio of Experimental/ Negative control (E/C)> 2 [31, 35]. Such values are marked with an asterisk (*).

Rv3620c secreted IFN-γ but the secretions of IL-5, IL1-10 and IL-17A were not detected in any group of mice immunized with the rDNA vaccine constructs (Fig 5).

The relative concentrations of Th1, Th2, Th17 and Treg cytokines secreted from spleen cells of mice in response to PPs, as calculated by the ratios of Th1:Th2 (IFN-γ:IL-5), Th1:Treg (IFN-γ:IL-10), and Th1:Th17 (IFN-γ:IL-17A), are shown in Fig 6. The analysis of these results

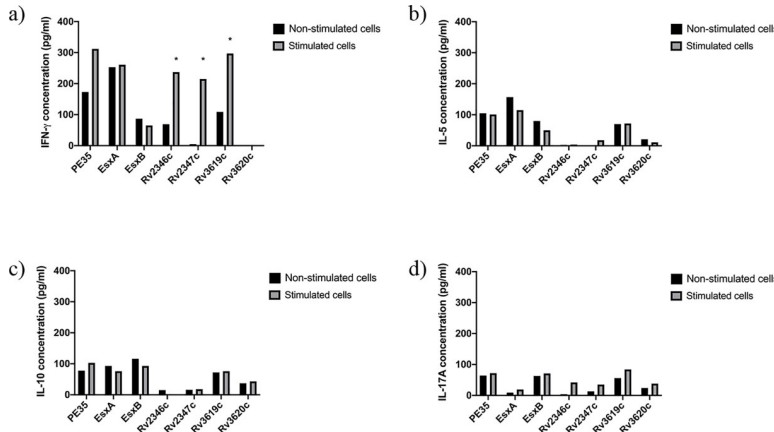

**Fig 3. Concentrations (pg/ml) of IFN-γ, IL-5, IL-10 and IL-17A in pools of culture supernatants of spleen cells obtained from mice (n = 5 per group) immunized with r*M. smegmatis*.** Spleen cells obtained from immunized mice were cultured in triplicates in the absence of any stimulant (negative control) and in the presence of stimulants (experimental), i.e. pools of peptides (PPs) covering the sequences of individual immunizing proteins. The supernatants were collected on day 6. The culture supernatants from triplicates of negative controls and each stimulant were pooled separately and tested for secreted a) IFN-γ, b) IL-5, c) IL-10, and d) IL-17A cytokines in duplicate wells of 96-well plates by ELISA. The average values of cytokine concentrations from duplicates are presented in the figure. The concentrations of cytokines in response to various stimulants were considered significant with quantities >100 pg/ml and the ratio of Experimental/ Negative control (E/C)> 2 [31, 35]. Such values are marked with an asterisk (*).

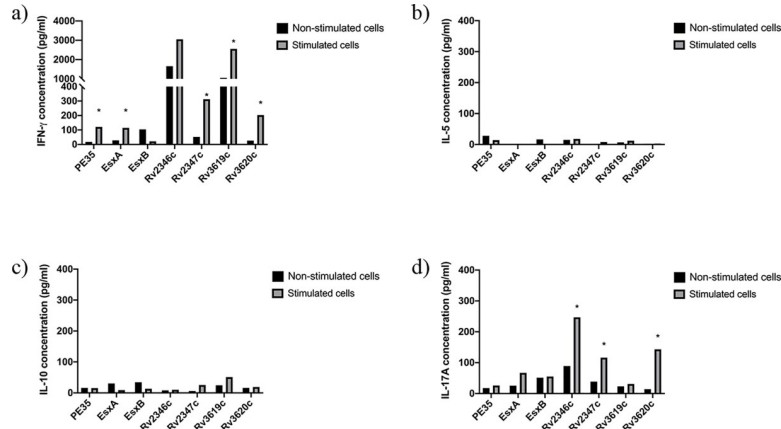

**Fig 4. Concentrations (pg/ml) of IFN-γ, IL-5, IL-10 and IL-17A in pools of culture supernatants of spleen cells obtained from mice (n = 5 per group) immunized with r*M. vaccae.*** Spleen cells obtained from immunized mice were cultured in triplicates in the absence of any stimulant (negative control) and in the presence of stimulants (experimental), i.e. pools of peptides (PPs) covering the sequences of individual immunizing proteins. The supernatants were collected on day 6. The culture supernatants from triplicates of negative controls and each stimulant were pooled separately and tested for secreted a) IFN-γ, b) IL-5, c) IL-10, and d) IL-17A cytokines in duplicate wells of 96-well plates by ELISA. The average values of cytokine concentrations from duplicates are presented in the figure. The concentrations of cytokines in response to various stimulants were considered significant with quantities >100 pg/ml and the ratio of Experimental/ Negative control (E/C)> 2 [31, 35]. Such values are marked with an asterisk (*).

for cytokine biases showed that Th1 biases, compared to Th2, Treg, Th17 or no biases, were more predominant with all adjuvants/delivery systems (Table 3). A similar analysis for the antigens showed that Rv2347c and Rv3619c always (15/15 possible combinations) induced Th1-biased responses with all adjuvants/delivery systems (Tables 3 and 4). Furthermore, Th1 responses were predominant with PE35, ESXA and Rv2346c (Table 4). The remaining two antigens, i.e. ESXB and Rv3620c induced predominantly non-Th1-baised responses (Table 4).

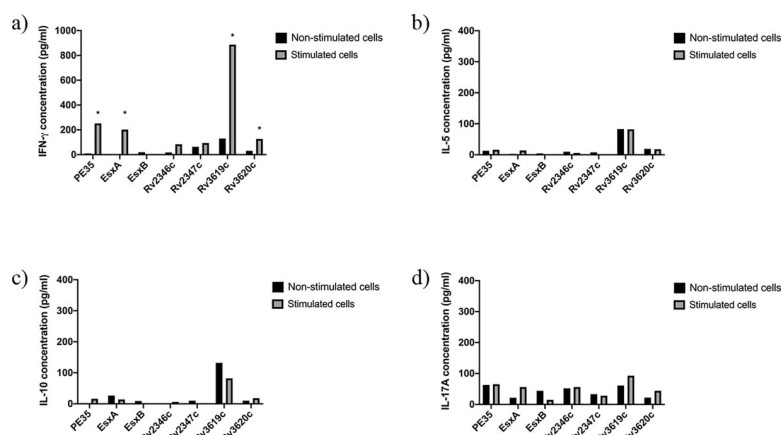

**Fig 5. Concentrations (pg/ml) of IFN-γ, IL-5, IL-10 and IL-17A in pools of culture supernatants of spleen cells obtained from mice (n = 5 per group) immunized with DNA vaccine constructs.** Spleen cells obtained from immunized mice were cultured in triplicates in the absence of any stimulant (negative control) and in the presence of stimulants (experimental), i.e. pools of peptides (PPs) covering the sequences of individual immunizing proteins. The supernatants were collected on day 6. The culture supernatants from triplicates of negative controls and each stimulant were pooled separately and tested for secreted a) IFN-γ, b) IL-5, c) IL-10, and d) IL-17A cytokines in duplicate wells of 96-well plates by ELISA. The average values of cytokine concentrations from duplicates are presented in the figure. The concentrations of cytokines in response to various stimulants were considered significant with quantities >100 pg/ml and the ratio of Experimental/ Negative control (E/C)> 2 [31, 35]. Such values are marked with an asterisk (*).

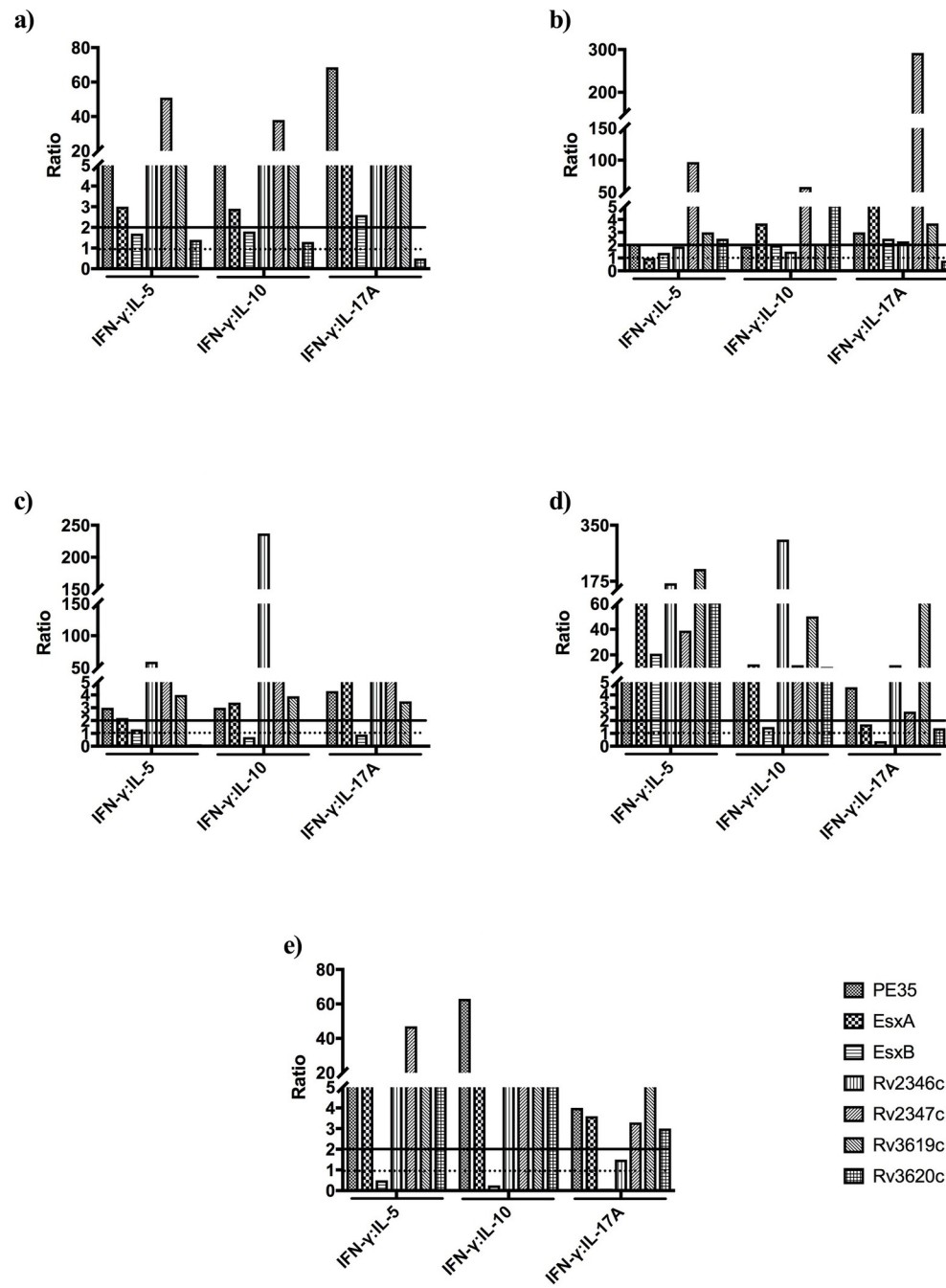

**Fig 6.** IFN-γ:IL-5, IFN-γ:IL-10 and IFN-γ:IL-17A ratios in the supernatants of spleen cells cultures in response to mixtures of peptides (PPs) from mice immunized with recombinant proteins, (a) emulsified with IFA, (b) adsorbed onto Alum, and (c) r*M. smegmatis*, (d) r*M. vaccae*, and (e) rDNA vaccine constructs. The ratios of IFN-γ:IL-5, IFN-γ:IL-10, and IFN-γ:IL-17A exceeding 2 (straight line) were considered as Th1 bias, 1 to 2 no bias, and <1 (dotted line) as Th2, Treg and Th17 biases, respectively.

## Discussion

Enhancement of the immunogenicity of *M. tuberculosis*-specific antigens may lead to the development of efficient subunit TB vaccines, as compared to the currently available BCG

**Table 3. T cells biases based on cytokine ratios (Th1:Th2, Th1:Treg and Th1:Th17) in the supernatants of spleen cell cultures in response to peptide pools from mice immunized with PE35, ESXA, ESXB, Rv2346c, Rv2347c, Rv3619c, and Rv3620c using various adjuvants/delivery systems.**

| Adjuvant/Delivery system | Cytokine ratio | T cell biases in response to peptide pools from mice immunized with | | | | | | |
|---|---|---|---|---|---|---|---|---|
| | | PE35 | ESXA | ESXB | Rv2346c | Rv2347c | Rv3619c | Rv3620c |
| IFA | Th1:Th2 | Th1 | Th1 | No bias | Th1 | Th1 | Th1 | No bias |
| | Th1:Treg | Th1 | Th1 | No bias | Th1 | Th1 | Th1 | No bias |
| | Th1:Th17 | Th1 | Th1 | Th1 | Th1 | Th1 | Th1 | Th17 |
| Alum | Th1:Th2 | Th1 | Th1 | No bias | No bias | Th1 | Th1 | Th1 |
| | Th1:Treg | No bias | Th1 | No bias | No bias | Th1 | Th1 | Th1 |
| | Th1:Th17 | Th1 | Th1 | Th1 | Th1 | Th1 | Th1 | Th17 |
| rM. smegmatis | Th1:Th2 | Th1 | Th1 | No bias | Th1 | Th1 | Th1 | Th2 |
| | Th1:Treg | Th1 | Th1 | Treg | Th1 | Th1 | Th1 | Treg |
| | Th1:Th17 | Th1 | Th1 | Th17 | Th1 | Th1 | Th1 | Th17 |
| rM. vaccae | Th1:Th2 | Th1 | Th1 | Th1 | Th1 | Th1 | Th1 | Th1 |
| | Th1:Treg | Th1 | Th1 | Treg | Th1 | Th1 | Th1 | Th1 |
| | Th1:Th17 | Th1 | No bias | Th17 | Th1 | Th1 | Th1 | No bias |
| DNA vaccine | Th1:Th2 | Th1 | Th1 | Th2 | Th1 | Th1 | Th1 | Th1 |
| | Th1:Treg | Th1 | Th1 | Treg | Th1 | Th1 | Th1 | Th1 |
| | Th1:Th17 | Th1 | Th1 | Th17 | No bias | Th1 | Th1 | Th1 |

vaccine. However, the delivery of these antigens requires appropriate adjuvants and delivery systems [26]. In this study, the effects of various adjuvants and delivery systems were evaluated in mice for immunogenicity of *M. tuberculosis*-specific proteins encoded by RD1 (PE35, EsxA and EsxB), RD7 (Rv2346c and Rv2347c), and RD9 (Rv3619c and Rv3620c). The spleen cells of immunized mice were tested for antigen-specific secretion of adaptive immune response-related Th1 (IFN-γ), Th2 (IL-5), Th17 (IL-17A) and Treg (IL-10) cytokines. The results show that with each delivery system, the protective Th1 cytokine IFN-γ was secreted in response to two or more antigens and Rv3619c induced a Th1 response with all adjuvants/delivery systems. ESXB was the only antigen which didn't induce antigen-specific Th1 response with any of the adjuvants/delivery systems and induced antigen-specific IL-5 when emulsified with Alum. On the other hand, immunization only with r*M. vaccae*/rv3619c resulted in the secretion of antigen-specific IL-17A. However, no antigen-specific IL-10 was secreted by any of the antigens with any adjuvant/delivery system. Interestingly, all antigens inducing IFN-γ secretion showed Th1 biases, when compared with Th2, Th17 and Treg responses. Previously, it has been shown that rBCG expressing ESXB antigen did not induce Th1 responses in BALB/c mice [31].

IFA is a water-in-oil emulsion that has been widely used in animal models since 1970s for evaluating the quality of vaccine candidates [36]. The suggested mechanism for the adjuvant effect of IFA is to prolong the duration of antigen persistence at the site of injection [37]. It has

**Table 4. A summary of T cell biases in the supernatant of spleen cells cultures in response to peptide pools from mice immunized with PE35, ESXA, ESXB, Rv2346c, Rv2347c, Rv3619c, and Rv3620c using various adjuvants/delivery systems.**

| | PE35 | ESXA | ESXB | Rv2346c | Rv2347c | Rv3619c | Rv3620c |
|---|---|---|---|---|---|---|---|
| Th1 bias | 14/15 | 14/15 | 3/15 | 12/15 | 15/15 | 15/15 | 7/15 |
| No bias | 1/15 | 1/15 | 5/15 | 3/15 | 0/15 | 0/15 | 3/15 |
| Th2 bias | 0/15 | 0/15 | 1/15 | 0/15 | 0/15 | 0/15 | 1/15 |
| Treg bias | 0/15 | 0/15 | 4/15 | 0/15 | 0/15 | 0/15 | 1/15 |
| Th17 bias | 0/15 | 0/15 | 2/15 | 0/15 | 0/15 | 0/15 | 3/15 |

been previously shown in mice, that administration of IFA alone induces the secretion of Th2 but not Th1 cytokines [38]. However, when emulsified with an antigen or DNA vaccine, IFA is a powerful adjuvant that induces a cellular rather than humoral immune response [38, 39]. In this study, purified recombinant proteins were emulsified with IFA in equal ratio (1:1) and mice were immunized intraperitoneally. The cytokine secretion from spleen cells, stimulated in vitro with the peptides of immunizing proteins, showed that mice immunized with purified recombinant proteins PE35, Rv2346c and Rv3619c emulsified with IFA, had increased secretion of protective IFN-γ with high Th1 biases, when compared with Th2, Treg and Th17 responses. Indeed, the Th1 cytokine secretion by spleen cells of these mice was the highest among all other delivery systems. Interestingly, however, these mice did not produce IL-5, IL-10 or IL-17A cytokines. Similar results have been reported by Xiang et al. [40]. In their experiments, mice were immunized with five ESX-fusion proteins (EsxB, EsxD, EsxG, EsxU and EsxM) emulsified with IFA and later infected with $10^7$ CFU of *M. bovis* BCG-Pasteur. The immunization with these proteins emulsified with IFA induced the secretion of protective Th1 cytokine (IFN-γ) from spleen cells and also lowered the bacterial load significantly in lungs [40].

Alum salts are known to be Th2-inducing adjuvants, however due to their safety and function of increasing the stability and immunogenicity of recombinant proteins, they have been widely used in pursuit of developing subunit TB vaccines and to compare new adjuvants [41–45]. Despite the positive correlation between the induction of high Th1 responses and protection against TB in mice, it has been shown that the use of Th2 inducing adjuvants, such as Alum, is not associated with harmful effects [46]. In fact, the secretion of certain degree of Th2 cytokines doesn't have an adverse consequence, as long as the Th1 response is substantial [47]. In our study, the spleen cells collected from immunized mice showed the induction of protective Th1 cytokine IFN-γ to Rv2347c and Rv3619c recombinant proteins, but their levels were lower than IFA group. Of interest, the secretions of Th2, Th17 and Treg cytokines were not detected in response to the peptide pools of Rv2347c and Rv3619c. In previously published studies, the immunization with immunodominant mycobacterial proteins emulsified with Alum had controversial outcomes. Orr and colleagues have shown that Alum in combination with glucopyranosyl lipid adjuvant (GLA) had a synergetic effect, promoted Th1 responses to ID93, and significantly reduced the bacterial load in the lungs and spleens of *M. tuberculosis*-infected mice [48]. On the other hand, Agger et al. showed that combining Ag85B-ESXA fusion protein with Alum and dimethyldioctadecylammonium (DDA) adjuvants did not reduce the bacterial load in lungs of infected mice [42, 46]. Besides, immunization of mice with the chimeric tuberculosis vaccine antigen H56 along with alum did not induce the secretion of primary T cell responses, rather it increased the primary B cell responses and IgG1 production [26].

The recombinant non-pathogenic mycobacteria expressing immunodominant *M. tuberculosis* antigens have been broadly used as TB vaccines in preclinical studies due to their rapid growth, genetic and structural homology to *M. tuberculosis* and induction of long-lasting immunity [49,50, 51, 52–53]. In addition, the safety of non-pathogenic mycobacteria has been well documented, and their efficacies have been studied as vaccines and immunotherapeutic agents for the treatment of TB and other diseases, e.g. cancers and autoimmune disorders [54, 55]. Specifically, *M. vaccae* has been approved in China as an immunotherapeutic agent to shorten the duration of TB drug therapy [55]. Besides, heat-killed *M. vaccae* is considered to be safe for HIV positive patients [56], and *M. smegmatis* is nontoxic in animal models lacking NK or T cells [57]. In this study, we also evaluated r*M. smegmatis* and r*M. vaccae* carrying a recombinant shuttle vector (rpDE22) as live vaccine candidates. The pDE22 shuttle vector contains a hygromycin-resistant gene marker, a hsp60 transcription signal, and the secretion

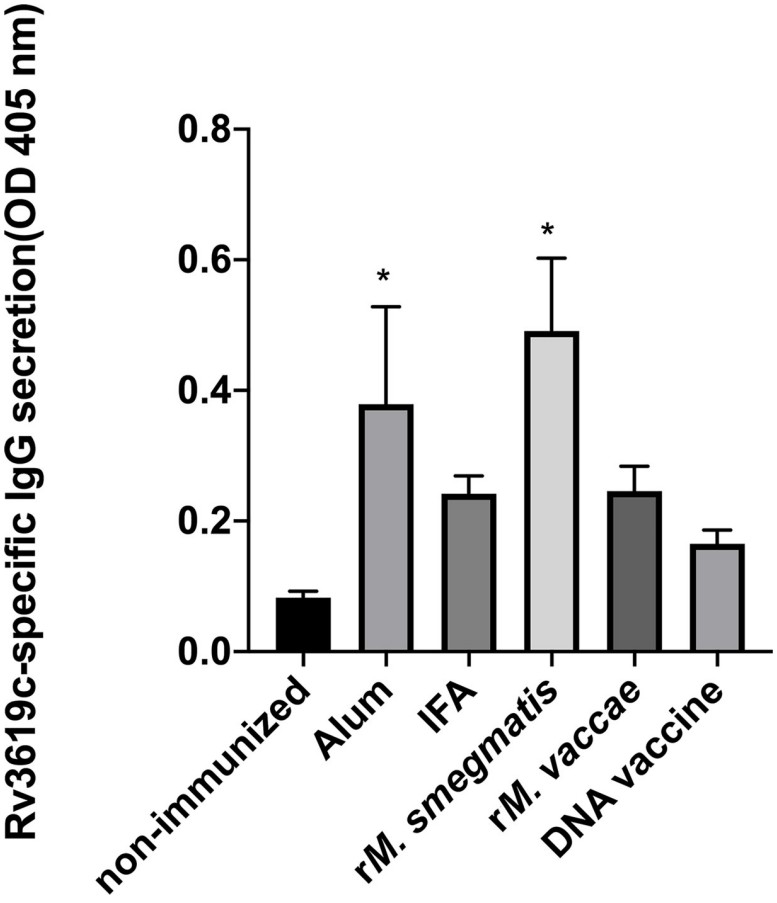

**Fig 7. Rv3619c-specific antibody reactivity in sera from groups of mice (n = 5 per group) immunized with Rv3619c in Alum, IFA, r*M. smegmatis*, r*M. vaccae*, and DNA vaccine construct.** Sera from non-immunized and immunized mice were tested for antibody reactivity by ELISA using the pure recombinant protein of Rv3619c as antigen according to standard procedures [63]. Data were obtained as optical density (OD) values measured at 405 nm. The statistical analysis of data was performed with a one-way analysis of variance (ANOVA) test followed by Bonferroni post hoc test for Rv3619c-specific IgG antibodies. The results were considered significant with $P < 0.05$ against non-immunized mice. Such values in the figure are marked with an asterisk (*).

signal of *M. tuberculosis* alpha antigen to export the expressed proteins into the extracellular milieu [23]. Our findings show that spleen cells obtained from mice immunized with r*M. smegmatis*/*rv2346*c, r*M. smegmatis*/*rv2347*c and r*M. smegmatis*/*rv3619c* induced the secretion of IFN-γ in response to the pools of synthetic peptides corresponding to these proteins, while IFN-γ was secreted only from spleen cells obtained from mice immunized with r*M. vaccae* expressing PE35, ESXA, Rv3619c, Rv3620c antigens.

DNA vaccines have been previously used to induce antigen specific cellular immunity and in immunotherapy against multidrug resistant TB in combination with anti-mycobacterial drugs in mice [58]. DNA vaccines are good adjuvant systems to induce the secretion of Th1 cytokines as well as Th1-like humoral immune response when injected intramuscularly [59]. The DNA vaccine plasmid used in this study was pUMVC6 which contains expression vector of CMV IE promoter at the 5' end and kanamycin marker. In addition, this expression vector contains human IL-2 secretory peptide, which acts as an adjuvant and allows the cloned gene to be secreted as a cytoplasmic protein to elicit immune responses in vivo [29]. It has previously been found that vaccinating mice with pUMVC6 constructs of PE35, ESXA, ESXB and

Rv3619c resulted in proliferation of spleen cells in vitro when stimulated with pools of peptides covering the sequence of the immunizing proteins [29]. A previous study has shown that immunization of mice with DNA vaccine construct pUMVC6/pe35 induced the secretion of Th1 cytokine IFN-γ but not Th2 and Treg cytokines IL-5 and IL-10, respectively [34]. Similarly, in this study, it was found that immunization of mice with DNA vaccine constructs expressing PE35, ESXA, Rv3619c and Rv3620c induced the secretion of Th1 cytokine, but not Th2, Th17 and Treg cytokines, in response to peptide pools of these antigens.

Our study shows that immunization with Rv3619c induced secretion of antigen-specific IFN-γ and Th1-biased responses with all adjuvants and delivery systems used, suggesting its appropriateness as a subunit vaccine against TB. However, some recent studies have shown that, in addition to Th1 responses, antibodies may also play a role in protection against TB [60, 61– 62]. We, therefore, determined the antigen-specific IgG antibody concentration in sera of mice immunized with Rv3619c with different adjuvants and delivery systems. The results showed that the concentrations of Rv3619c-specifc antibodies were higher in sera of all groups of immunized mice when compared with non-immunized mice, but significantly higher concentrations (P <0.05) were observed in mice immunized with Rv3619c in Alum and r*M. smegmatis* (Fig 7).

The antigen-specific Th1 cytokine and antibody responses suggest that Rv3619c may be useful as a subunit vaccine against TB. However, measuring only immune responses, in this study, is a limitation due to lack of immune correlates of protection in TB [64].

## Conclusions

In this study, seven immunodominant *M. tuberculosis*-specific antigens were obtained in recombinant forms, expressed in non-pathogenic *M. smegmatis* and *M. vaccae* and cloned in a DNA vaccine vector. The recombinants, along with chemical adjuvants and delivery systems, were evaluated in mice for the antigen-specific secretion of protective Th1 and Th17 as well as pathologic Th2, Th17 and Treg cytokines. The results show that all antigens, except ESXB, induced the secretion of the protective Th1 cytokine IFN-γ with two or more delivery systems. However, Rv3619c was the only antigen that showed consistent secretions of significant concentrations of Th1 cytokine IFN-γ and Th1 biases with all adjuvants and delivery systems. Immunization with Rv3619c also induced antigen-specific IgG antibodies with significantly higher concentrations in mice immunized with Rv3619c + Alum and *rM. smegmatis*. These results suggest that the quality and type of immune response depend upon the antigens as well as the adjuvants/delivery systems used.

## Supporting information

**S1 Fig. Cytokine secretion from spleen cells stimulated with ConA.** Concentrations (pg/ml) of IFN-γ, IL-5, IL-10 and IL-17A in pools of culture supernatants of spleen cells obtained from mice (n = 5 per group) immunized with purified recombinant proteins using a) Alum, b) IFA, c) r*M. smegmatis*, d) r*M. vaccae*, and e) DNA vaccine constructs. Spleen cells obtained from immunized mice were cultured in triplicates in the absence of any stimulant (negative control) and in the presence of stimulant (experimental), i.e. Con A. The supernatants were collected on day 3. The culture supernatants from triplicates of negative controls and the stimulant were pooled separately and tested for secreted IFN-γ, IL-5, IL-10 and IL-17A cytokines in duplicate wells of 96-well plates by ELISA. The average values of cytokine concentrations from duplicates are presented in the Figure.
(TIF)

## Acknowledgments

We thank Fatima Hussain for technical assistance.

## Author Contributions

**Conceptualization:** Hussain A. Safar, Abu Salim Mustafa, Hanady A. Amoudy, Ahmed El-Hashim.

**Data curation:** Hussain A. Safar, Abu Salim Mustafa.

**Formal analysis:** Hussain A. Safar, Abu Salim Mustafa, Hanady A. Amoudy, Ahmed El-Hashim.

**Funding acquisition:** Hussain A. Safar, Abu Salim Mustafa, Hanady A. Amoudy, Ahmed El-Hashim.

**Investigation:** Hussain A. Safar, Abu Salim Mustafa, Hanady A. Amoudy, Ahmed El-Hashim.

**Methodology:** Hussain A. Safar, Abu Salim Mustafa, Hanady A. Amoudy.

**Project administration:** Abu Salim Mustafa.

**Resources:** Abu Salim Mustafa.

**Software:** Hussain A. Safar.

**Supervision:** Abu Salim Mustafa, Hanady A. Amoudy, Ahmed El-Hashim.

**Validation:** Hussain A. Safar, Abu Salim Mustafa, Hanady A. Amoudy, Ahmed El-Hashim.

**Visualization:** Hussain A. Safar, Abu Salim Mustafa, Hanady A. Amoudy, Ahmed El-Hashim.

**Writing – original draft:** Hussain A. Safar.

**Writing – review & editing:** Abu Salim Mustafa, Hanady A. Amoudy, Ahmed El-Hashim.

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
