## [Decision Letter · Decision Letter 0]

25 Jun 2019

PONE-D-19-14947

The effect of adjuvants and delivery systems on Th1, Th2, Th17 and Treg cytokine responses in mice immunized with Mycobacterium tuberculosis-specific proteins

PLOS ONE

Dear Dr. Mustafa,

Thank you for submitting your manuscript to PLOS ONE. After careful consideration, we feel that it has merit but does not fully meet PLOS ONE’s publication criteria as it currently stands. Therefore, we invite you to submit a revised version of the manuscript that addresses the points raised during the review process.

A major limitation of the manuscript is the fact that the vaccines were compared only by measuring immune responses without evidence of protection. As there is no accepted immune correlate of protection for TB vaccines it is not feasible to identify promising candidates on the basis of immunogenicity. If protection data are available, these should be presented in this manuscript in order to support the conclusions that one antigen was superior to the others. Indeed Reviewer 1 has questioned this interpretation of superiority of Rv3619c based upon the data as presented. If protection data are not available, the limitations of the immune response data and their interpretation must be clearly stated in the Discussion. In addition, the discussion must be changed where the immune response data in this manuscript are compared / contrasted to published protection data – such comparisons cannot be made.

Several other issues should be addressed, as identified by the reviewers. In particular:

Presentation of the data in graphical format, rather than tables and the application of statistical methods. Currently the terminology used is ‘considered significant’ and no explanation is given for what this actually means with regards to statistics.Description/discussion of studies involving Alum. As mentioned by Reviewer 1, Alum is not a relevant adjuvant for TB vaccines and the sentence in lines 320-322 stating that Alum salts are widely used for TB vaccines is not correct. Furthermore, this statement is not supported by Ref.41 which describes correlates of protection following vaccination with BCG vaccines and has no mention of Alum. The discussion about the small number of studies using Alum is therefore redundant.Reviewer 2 has raised an important point about endotoxin levels which should be clarified.Comment and provide data if available on antibody-mediated responses and responses in the lungs (reviewer 2 and 1 respectively).Provide a rationale for using BALB/c mice and alter the discussion to take into account that differences between studies could be attributable to the strain of mice (reviewer 2).

We would appreciate receiving your revised manuscript by Aug 09 2019 11:59PM. To enhance the reproducibility of your results, we recommend that if applicable you deposit your laboratory protocols in protocols.io, where a protocol can be assigned its own identifier (DOI) such that it can be cited independently in the future. For instructions see: http://journals.plos.org/plosone/s/submission-guidelines#loc-laboratory-protocols

We look forward to receiving your revised manuscript.

Kind regards,

Ann Rawkins, PhD

Academic Editor

PLOS ONE

Journal Requirements:

Reviewers' comments:

Reviewer's Responses to Questions

**Comments to the Author**

1. Is the manuscript technically sound, and do the data support the conclusions?

Reviewer #1: No

Reviewer #2: Partly

2. Has the statistical analysis been performed appropriately and rigorously? 

Reviewer #1: No

Reviewer #2: No

3. Have the authors made all data underlying the findings in their manuscript fully available?

Reviewer #1: Yes

Reviewer #2: Yes

4. Is the manuscript presented in an intelligible fashion and written in standard English?

Reviewer #1: No

Reviewer #2: Yes

5. Review Comments to the Author

Reviewer #1: Major Comments:

The manuscript titled "The effect of adjuvants and delivery systems on Th1, Th2, Th17 and Treg cytokine responses in mice immunized with Mycobacterium tuberculosis-specific proteins" is an effort in the direction of elucidating immune responses induced by several Mycobacterium tuberculosis-specific antigens.

Authors concluded that Rv3619c (EsxV, ESAT-6-like protein) has a potential as a vaccine antigen because of its Th1-inducing ability. However, the authors jumped to a conclusion without any evidence for protection efficacy testing. In addition, although Th1-type T-cell response is crucial for the protective immunity, other T-cell responses are also important for the protection against tuberculosis.

Authors included ESXA (ESAT-6) in this study. I do not understand how Rv3619c induced a more robust Th1 response than ESXA. Also, Rv2347c showed Th1-biased response in all experiments like Rv3619c.

I suggest an extensive review across the manuscript and should focus on the immunogenicity, not construction of expression vectors. Additionally, I do not think alum adjuvant is proper for tuberculosis vaccine. Thus, I would like to recommend the exclusion of data for alum adjuvant for the readers.

I do not understand why the authors did not investigate immune responses in the lungs following immunizations.

Minor Comments:

Lines 149-150: Gene name should be italic. Throughout the manuscript, please check them.

Lines 200-203: Although the authors cited reference #17 to calculate the ratios of Th-polarized responses, is this method is approved and applicable to mice?

Please check the abbreviations for tuberculosis, Mycobacterium tuberculosis, etc.. Line 323: TB, Line 336: MTB, Line 339: tuberculosis

The resolution for all figures should be high. Some figures are unreadable.

Reviewer #2: This manuscript describes the evaluation of the immunogenicity of a number of M.tb antigens presented in regions that are absent from BCG.

These antigens were delivered as proteins with adjuvants, DNA or recombinant mycobacteria. Systemic immune responses were divided into the induction of Th1, Th2 or Th17.

The authors presented an impressive amount of work however I have some major comments:

- This manuscript could have really benefited from protection studies to evaluate the efficacy of the different vaccine candidates. Although I appreciate that it would have been impossible to test all vaccine combinations, it would have been important to at least test the efficacy of the vaccination regime that the authors thought was the most superior based on immunology. Such experiment could have validated the immunogenicity data generated in this study. The lack of immune correlates of protection is a critical limitation in the TB vaccine development field and therefore down-selecting vaccine candidates on the basis of just immune responses might not be the most accurate approach. The authors should explain their decision to not perform protection experiments. The above limitations should be emphasized in the discussion.

- Have the authors tested for the presence of endotoxin in their proteins? If yes the level should be specified in the methodology section

- Although the role of antibodies is still not well understood there is an increasing body of evidence to suggest that they might have an important protective role. Why were antibody responses not measured in these experiments? Discussion should be amended to include some of this information

- All the immunological data are presented in the form of tables rather than graphs. It would be useful for at least some of the data to be presented in a graphical format. Although the positive responses to ConA are important to demonstrate the viability and responsive capacity of cells, they to distract from the main data. As a result it might be better to have the ConA responses as supporting tables.

- No statistical testing was performed on any of the data.

- Can the authors explain why they chose BALB/c mice for their experiments? The study that the authors mention in their discussion (line 305) used C57BL/6 mice. Could that have potentially affected the Th1/Th2 responses knowing that the two mouse strains have a different Th bias? Perhaps discussion should be expanded to include these points.

Minor:

- The quality of figure 3 is really low. Would it be possible to replace with higher resolution pictures?

- BCG – abbreviation should be explained

- line 286 in discussion. What do the authors mean by safer? Subunit vaccines will have to be administered with or as a boost to BCG. How does that make them safer compared to BCG alone?

- Line 361 Can the authors clarify which other stain do they compare the C57BL/6 mice to? And can they provide a reference to support this statement?

6. PLOS authors have the option to publish the peer review history of their article (what does this mean?). If published, this will include your full peer review and any attached files.

Reviewer #1: No

Reviewer #2: No

---

## [Author Response · Author response to Decision Letter 0]

17 Dec 2019

'Response to Reviewers'

Editor’s comments:

General comments

A major limitation of the manuscript is the fact that the vaccines were compared only by measuring immune responses without evidence of protection. As there is no accepted immune correlate of protection for TB vaccines it is not feasible to identify promising candidates on the basis of immunogenicity. If protection data are available, these should be presented in this manuscript in order to support the conclusions that one antigen was superior to the others. Indeed, Reviewer 1 has questioned this interpretation of superiority of Rv3619c based upon the data as presented. If protection data are not available, the limitations of the immune response data and their interpretation must be clearly stated in the Discussion. In addition, the discussion must be changed where the immune response data in this manuscript are compared / contrasted to published protection data – such comparisons cannot be made.

Response: The protection data are not available from this study. However, a previous study (Ansari et al. Reference no. 56 in the revised manuscript) has shown the protective efficacy of Rv3619c in the same mouse strain which was used by us, i.e. BALB/c. As suggested by the Editor, the discussion has been changed where the immune response data were compared/contrasted with the protection data.

Specific comments:

Comment: Presentation of the data in graphical format, rather than tables and the application of statistical methods. Currently the terminology used is ‘considered significant’ and no explanation is given for what this actually means with regards to statistics.

Response: The data has been presented in graphical format, as suggested. The information on the use of statistics and the criteria for significance has been added on page 12 of the revised manuscript.

Comment: Description/discussion of studies involving Alum. As mentioned by Reviewer 1, Alum is not a relevant adjuvant for TB vaccines and the sentence in lines 320-322 stating that Alum salts are widely used for TB vaccines is not correct. Furthermore, this statement is not supported by Ref.41 which describes correlates of protection following vaccination with BCG vaccines and has no mention of Alum. The discussion about the small number of studies using Alum is therefore redundant.

Response: Alum, being a Th2 adjuvant, provides the baseline to compare with other adjuvants. Hence, we would like to retain the results obtained with this adjuvant. Furthermore, for the same reason, Alum has been used as an adjuvant in TB vaccine research. A list of 14 such references from PubMed is given below.

1. Mani R, Gupta M, Malik A, Tandon R, Prasad R, Bhatnagar R, Banerjee N. Adjuvant Potential of Poly-α-l-Glutamine from the Cell Wall of Mycobacterium tuberculosis. Infect Immun. 2018 Sep 21;86(10).

2. Tang J, Sun M, Shi G, Xu Y, Han Y, Li X, Dong W, Zhan L, Qin C. Toll-Like Receptor 8 Agonist Strengthens the Protective Efficacy of ESAT-6 Immunization to Mycobacterium tuberculosis Infection. Front Immunol. 2018 Jan 24;8:1972.

3. Tirado Y, Puig A, Alvarez N, Borrero R, Aguilar A, Camacho F, Reyes F, Fernandez S, Perez JL, Acevedo R, Mata Espinoza D, Payan JA, Garcia ML, Kadir R, Sarmiento ME, Hernandez-Pando R, Norazmi MN, Acosta A. Mycobacterium smegmatis proteoliposome induce protection in a murine progressive pulmonary tuberculosis model. Tuberculosis (Edinb). 2016 Dec;101:44-48.

4. Knudsen NP, Olsen A, Buonsanti C, Follmann F, Zhang Y, Coler RN, Fox CB, Meinke A, D'Oro U, Casini D, Bonci A, Billeskov R, De Gregorio E, Rappuoli R, Harandi AM, Andersen P, Agger EM. Different human vaccine adjuvants promote distinct antigen-independent immunological signatures tailored to different pathogens. Sci Rep. 2016 Jan 21;6:19570. 

5. Ciabattini A, Pettini E, Fiorino F, Pastore G, Andersen P, Pozzi G, Medaglini D. Modulation of Primary Immune Response by Different Vaccine Adjuvants. Front Immunol. 2016 Oct 17;7:427. 

6. Sun T, Han H, Hudalla GA, Wen Y, Pompano RR, Collier JH. Thermal stability of self-assembled peptide vaccine materials. Acta Biomater. 2016 Jan;30:62-71.

7. Tirado Y, Puig A, Alvarez N, Borrero R, Aguilar A, Camacho F, Reyes F, Fernández S, Pérez JL, Espinoza DM, Payán JA, Sarmiento ME, Norazmi MN, Hernández-Pando R, Acosta A. Protective capacity of proteoliposomes from Mycobacterium bovis BCG in a mouse model of tuberculosis. Hum Vaccin Immunother. 2015;11(3):657-61.

8. García Mde L, Borrero R, Lanio ME, Tirado Y, Alvarez N, Puig A, Aguilar A, Canet L, Mata Espinoza D, Barrios Payán J, Sarmiento ME, Hernández-Pando R, Norazmi MN, Acosta A. Protective effect of a lipid-based preparation from Mycobacterium smegmatis in a murine model of progressive pulmonary tuberculosis. Biomed Res Int. 2014;2014:273129. 

9. Orr MT, Fox CB, Baldwin SL, Sivananthan SJ, Lucas E, Lin S, Phan T, Moon JJ, Vedvick TS, Reed SG, Coler RN. Adjuvant formulation structure and composition are critical for the development of an effective vaccine against tuberculosis. J Control Release. 2013 Nov 28;172(1):190-200. 

10. Rodriguez L, Tirado Y, Reyes F, Puig A, Kadir R, Borrero R, Fernandez S, Reyes G, Alvarez N, Garcia MA, Sarmiento ME, Norazmi MN, Perez Quinoy JL, Acosta A. Proteoliposomes from Mycobacterium smegmatis induce immune cross-reactivity against Mycobacterium tuberculosis antigens in mice. Vaccine. 2011 Aug 26;29(37):6236-41.

11. Chen L, Xu M, Wang ZY, Chen BW, Du WX, Su C, Shen XB, Zhao AH, Dong N, Wang YJ, Wang GZ. The development and preliminary evaluation of a new Mycobacterium tuberculosis vaccine comprising Ag85b, HspX and CFP-10:ESAT-6 fusion protein with CpG DNA and aluminum hydroxide adjuvants. FEMS Immunol Med Microbiol. 2010 Jun 1;59(1):42-52.

12. Kamath AT, Valenti MP, Rochat AF, Agger EM, Lingnau K, von Gabain A, Andersen P, Lambert PH, Siegrist CA. Protective anti-mycobacterial T cell responses through exquisite in vivo activation of vaccine-targeted dendritic cells. Eur J Immunol. 2008 May;38(5):1247-56. 

13. Bosio CM, Orme IM. Effective, nonsensitizing vaccination with culture filtrate proteins against virulent Mycobacterium bovis infections in mice. Infect Immun. 1998 Oct;66(10):5048-51.

14. Lindblad EB, Elhay MJ, Silva R, Appelberg R, Andersen P. Adjuvant modulation of immune responses to tuberculosis subunit vaccines. Infect Immun. 1997 Feb;65(2):623-9.

The previous reference 41 has been replaced with new references 41 to 45.

Comment: Reviewer 2 has raised an important point about endotoxin levels which should be clarified.

Response: The endotoxin levels in the purified recombinant antigen preparations were not quantified but we expect that the concentrations would have been quite low because two different types of affinity columns were used for the purifications of recombinant proteins. 

Comment: Comment and provide data if available on antibody-mediated responses and responses in the lungs (reviewer 2 and 1 respectively).

Response: The antibody responses to Rv3619c are added in the revised manuscript in discussion. The immunizations were not performed through nasal route, hence immune responses in the lungs were not measured.

Comment: Provide a rationale for using BALB/c mice and alter the discussion to take into account that differences between studies could be attributable to the strain of mice (reviewer 2).

Response: BALB/c mice have been extensively used in TB immunology/vaccine research. Some of the papers published in 2019 and available in PubMed are given below. 

1. Broset E, Saubi N, Guitart N, Aguilo N, Uranga S, Kilpeläinen A, Eto Y, Hanke T, Gonzalo-Asensio J, Martín C, Joseph-Munné J. MTBVAC-Based TB-HIV Vaccine Is Safe, Elicits HIV-T Cell Responses, and Protects against Mycobacterium tuberculosis in Mice. Mol Ther Methods Clin Dev. 2019; 13:253-264.

2. Xiao TY, Liu HC, Li XQ, Huang MX, Li GL, Li N, Yan YH, Luo Q, Wang XZ, Li MC, Wan KL. Immunological Evaluation of a Novel Mycobacterium tuberculosis Antigen Rv0674. Biomed Environ Sci. 2019; 32(6):427-437.

3. Nagpal PS, Kesarwani A, Sahu P, Upadhyay P. Aerosol immunization by alginate coated mycobacterium (BCG/MIP) particles provide enhanced immune response and protective efficacy than aerosol of plain mycobacterium against M.tb. H37Rv infection in mice.

BMC Infect Dis. 2019; 19(1):568. 

4. Komine-Aizawa S, Jiang J, Mizuno S, Hayakawa S, Matsuo K, Boyd LF, Margulies DH, Honda M. MHC-restricted Ag85B-specific CD8+ T cells are enhanced by recombinant BCG prime and DNA boost immunization in mice. Eur J Immunol. 2019;49(9):1399-1414.

5. Moreno-Mendieta S, Barrera-Rosales A, Mata-Espinosa D, Barrios-Payán J, Sánchez S, Hernández-Pando R, Rodríguez-Sanoja R. Raw starch microparticles as BCG adjuvant: Their efficacy depends on the virulence of the infection strains.

Vaccine. 2019;37(38):5731-5737. 

6. Fitzpatrick M, Ho MM, Clark S, Dagg B, Khatri B, Lanni F, Williams A, Brennan M, Laddy D, Walker B. Comparison of pellicle and shake flask-grown BCG strains by quality control assays and protection studies. Tuberculosis (Edinb). 2019;114:47-53. 

7. Sawutdeechaikul P, Cia F, Bancroft G, Wanichwecharungruang S, Sittplangkoon C, Palaga T. Oxidized Carbon Nanosphere-Based Subunit Vaccine Delivery System Elicited Robust Th1 and Cytotoxic T Cell Responses. J Microbiol Biotechnol. 2019; 29(3):489-499. 

8. Okay S, Çetin R, Karabulut F, Doğan C, Sürücüoğlu S, Kızıldoğan AK.

Immune responses elicited by the recombinant Erp, HspR, LppX, MmaA4, and OmpA proteins from Mycobacterium tuberculosis in mice. Acta Microbiol Immunol Hung. 2019; 66(2):219-234.

9. Bull NC, Stylianou E, Kaveh DA, Pinpathomrat N, Pasricha J, Harrington-Kandt R, Garcia-Pelayo MC, Hogarth PJ, McShane H.

Enhanced protection conferred by mucosal BCG vaccination associates with presence of antigen-specific lung tissue-resident PD-1+ KLRG1- CD4+ T cells. Mucosal Immunol. 2019; 12(2):555-564. 

As suggested by the Editor, the discussion has been altered to take into account that differences between studies could be attributable to the strain of mice.

Reviewer #1: Major Comments:

Comment:

Authors concluded that Rv3619c (EsxV, ESAT-6-like protein) has a potential as a vaccine antigen because of its Th1-inducing ability. However, the authors jumped to a conclusion without any evidence for protection efficacy testing. In addition, although Th1-type T-cell response is crucial for the protective immunity, other T-cell responses are also important for the protection against tuberculosis.

Response:

The antigen RV3619c was shown to be protective in BALB/c mouse model of TB, as given in reference 56 (Ansari et al. PLoS One. 2011;6(8):e22889). Our correlation with protection is based on that study. We thank the reviewer for stating that Th1-type T-cell responses are crucial for the protective immunity against tuberculosis. Furthermore, we agree with the reviewer that other T-cell responses are also important for the protection against tuberculosis. Therefore, we have added antibody responses (primarily Th-2) to Rv3619c in the revised manuscript.

Comment: Authors included ESXA (ESAT-6) in this study. I do not understand how Rv3619c induced a more robust Th1 response than ESXA. Also, Rv2347c showed Th1-biased response in all experiments like Rv3619c.

Response: Based on statistical analysis of data, we have modified the results and conclusions. Even with the statistical analysis, Rv3619c was found superior than ESXA (Fig. 7 of the revised manuscript). One of the reasons could be the ability of ESXA (ESAT6) to activate regulatory T (Treg) cells, as reported in the references given below. 

Feruglio SL, Kvale D, Dyrhol-Riise AM. T cell responses and regulation and the impact of in vitro IL‐10 and TGF‐β modulation during treatment of active tuberculosis. Scand J Immunol. 2017; 85(2):138-146. 

Wu YE, Du ZR, Cai YM, Peng WG, Zheng GZ, Zheng GL, Wu LB, Li K. Effective expansion of forkhead box P3⁺ regulatory T cells via early secreted antigenic target 6 and antigen 85 complex B from Mycobacterium tuberculosis. Mol Med Rep. 2015; 11(4):3134-42.

Jackson-Sillah D, Cliff JM, Mensah GI, Dickson E, Sowah S, Tetteh JK, Addo KK, Ottenhoff TH, Bothamley G, Dockrell HM. Recombinant ESAT-6-CFP10 fusion protein induction of Th1/Th2 cytokines and FoxP3 expressing Treg cells in pulmonary TB. PLoS One. 2013; 8(6):e68121.

Comment: I suggest an extensive review across the manuscript and should focus on the immunogenicity, not construction of expression vectors. Additionally, I do not think alum adjuvant is proper for tuberculosis vaccine. Thus, I would like to recommend the exclusion of data for alum adjuvant for the readers.

Response: As suggested by the reviewer, the revised manuscript focuesses on immunogenicity and the results related to the construction of expression vectors have been deleted. Alum, being a Th2 adjuvant, provides the baseline to compare with other adjuvants. Hence, we would like to retain the results obtained with this adjuvant. Furthermore, for the same reason, Alum has been used as an adjuvant in TB vaccine research. A list of 14 such references from PubMed is given below.

1. Mani R, Gupta M, Malik A, Tandon R, Prasad R, Bhatnagar R, Banerjee N. Adjuvant Potential of Poly-α-l-Glutamine from the Cell Wall of Mycobacterium tuberculosis. Infect Immun. 2018 Sep 21;86(10).

2. Tang J, Sun M, Shi G, Xu Y, Han Y, Li X, Dong W, Zhan L, Qin C.Toll-Like Receptor 8 Agonist Strengthens the Protective Efficacy of ESAT-6 Immunization to Mycobacterium tuberculosis Infection. Front Immunol. 2018 Jan 24;8:1972.

3. Tirado Y, Puig A, Alvarez N, Borrero R, Aguilar A, Camacho F, Reyes F, Fernandez S, Perez JL, Acevedo R, Mata Espinoza D, Payan JA, Garcia ML, Kadir R, Sarmiento ME, Hernandez-Pando R, Norazmi MN, Acosta A. Mycobacterium smegmatis proteoliposome induce protection in a murine progressive pulmonary tuberculosis model. Tuberculosis (Edinb). 2016 Dec;101:44-48.

4. Knudsen NP, Olsen A, Buonsanti C, Follmann F, Zhang Y, Coler RN, Fox CB, Meinke A, D'Oro U, Casini D, Bonci A, Billeskov R, De Gregorio E, Rappuoli R, Harandi AM, Andersen P, Agger EM. Different human vaccine adjuvants promote distinct antigen-independent immunological signatures tailored to different pathogens. Sci Rep. 2016 Jan 21;6:19570. 

5. Ciabattini A, Pettini E, Fiorino F, Pastore G, Andersen P, Pozzi G, Medaglini D. Modulation of Primary Immune Response by Different Vaccine Adjuvants. Front Immunol. 2016 Oct 17;7:427. 

6. Sun T, Han H, Hudalla GA, Wen Y, Pompano RR, Collier JH. Thermal stability of self-assembled peptide vaccine materials. Acta Biomater. 2016 Jan;30:62-71.

7. Tirado Y, Puig A, Alvarez N, Borrero R, Aguilar A, Camacho F, Reyes F, Fernández S, Pérez JL, Espinoza DM, Payán JA, Sarmiento ME, Norazmi MN, Hernández-Pando R, Acosta A. Protective capacity of proteoliposomes from Mycobacterium bovis BCG in a mouse model of tuberculosis. Hum Vaccin Immunother. 2015;11(3):657-61.

8. García Mde L, Borrero R, Lanio ME, Tirado Y, Alvarez N, Puig A, Aguilar A, Canet L, Mata Espinoza D, Barrios Payán J, Sarmiento ME, Hernández-Pando R, Norazmi MN, Acosta A. Protective effect of a lipid-based preparation from Mycobacterium smegmatis in a murine model of progressive pulmonary tuberculosis. Biomed Res Int. 2014;2014:273129. 

9. Orr MT, Fox CB, Baldwin SL, Sivananthan SJ, Lucas E, Lin S, Phan T, Moon JJ, Vedvick TS, Reed SG, Coler RN. Adjuvant formulation structure and composition are critical for the development of an effective vaccine against tuberculosis. J Control Release. 2013 Nov 28;172(1):190-200. 

10. Rodriguez L, Tirado Y, Reyes F, Puig A, Kadir R, Borrero R, Fernandez S, Reyes G, Alvarez N, Garcia MA, Sarmiento ME, Norazmi MN, Perez Quinoy JL, Acosta A. Proteoliposomes from Mycobacterium smegmatis induce immune cross-reactivity against Mycobacterium tuberculosis antigens in mice. Vaccine. 2011 Aug 26;29(37):6236-41.

11. Chen L, Xu M, Wang ZY, Chen BW, Du WX, Su C, Shen XB, Zhao AH, Dong N, Wang YJ, Wang GZ. The development and preliminary evaluation of a new Mycobacterium tuberculosis vaccine comprising Ag85b, HspX and CFP-10:ESAT-6 fusion protein with CpG DNA and aluminum hydroxide adjuvants. FEMS Immunol Med Microbiol. 2010 Jun 1;59(1):42-52.

12. Kamath AT, Valenti MP, Rochat AF, Agger EM, Lingnau K, von Gabain A, Andersen P, Lambert PH, Siegrist CA. Protective anti-mycobacterial T cell responses through exquisite in vivo activation of vaccine-targeted dendritic cells. Eur J Immunol. 2008 May;38(5):1247-56. 

13. Bosio CM, Orme IM. Effective, nonsensitizing vaccination with culture filtrate proteins against virulent Mycobacterium bovis infections in mice. Infect Immun. 1998 Oct;66(10):5048-51.

14. Lindblad EB, Elhay MJ, Silva R, Appelberg R, Andersen P. Adjuvant modulation of immune responses to tuberculosis subunit vaccines. Infect Immun. 1997 Feb;65(2):623-9.

Comment: I do not understand why the authors did not investigate immune responses in the lungs following immunizations.

Response: The immune responses in the lungs were not investigated because the immunizations were systemic and not intra-nasal.

Minor Comments:

Lines 149-150: Gene name should be italic. Throughout the manuscript, please check them.

Response: Gene names have been given in italics, throughout the manuscript.

Comment: Please check the abbreviations for tuberculosis, Mycobacterium tuberculosis, etc.. Line 323: TB, Line 336: MTB, Line 339: tuberculosis

Response: The abbreviation for Mycobacterium tuberculosis is consistently changed from MTB to M. tuberculosis and tuberculosis to TB.

Comment: The resolution for all figures should be high. Some figures are unreadable.

Response: The resolution of all figures has been upgraded to meet the PLOS ONE standard.

Reviewer #2:

Comment: This manuscript could have really benefited from protection studies to evaluate the efficacy of the different vaccine candidates. Although I appreciate that it would have been impossible to test all vaccine combinations, it would have been important to at least test the efficacy of the vaccination regime that the authors thought was the most superior based on immunology. Such experiment could have validated the immunogenicity data generated in this study. The lack of immune correlates of protection is a critical limitation in the TB vaccine development field and therefore down-selecting vaccine candidates on the basis of just immune responses might not be the most accurate approach. The authors should explain their decision to not perform protection experiments. The above limitations should be emphasized in the discussion.

Response: The aim of this work was to study the immunogenicity of seven M. tuberculosis-specific antigens in BALB/C mice using different adjuvants and delivery systems. The protection efficacy of the antigen, found best in this study, has already been demonstrated earlier in BALB/c mice (Reference 57 of the first submitted manuscript). Therefore, we didn’t repeat the protection experiments in this study. In the revised manuscript, we have focused on immunogenicity and deleted the overemphasis on protection.

Comment: Have the authors tested for the presence of endotoxin in their proteins? If yes the level should be specified in the methodology section.

Response: The presence of endotoxin was not tested in the purified recombinant proteins because these were purified to homogeneity using two types of affinity columns. 

Comment: Although the role of antibodies is still not well understood there is an increasing body of evidence to suggest that they might have an important protective role. Why were antibody responses not measured in these experiments? Discussion should be amended to include some of this information

Response: The aim of this study was to determine the effect of adjuvants and delivery systems on cytokine responses. Hence, the antibody responses were not studied. However, in response to comment of this reviewer, antibody responses to Rv3619c are added in the discussion section of the revised manuscript. 

Comment: All the immunological data are presented in the form of tables rather than graphs. It would be useful for at least some of the data to be presented in a graphical format. Although the positive responses to ConA are important to demonstrate the viability and responsive capacity of cells, they to distract from the main data. As a result, it might be better to have the ConA responses as supporting tables.

Response: As suggested by the reviewer, some immunological data are presented in graphical format and the ConA Responses have been moved to a supporting figure.

Comment: No statistical testing was performed on any of the data.

Response: The statistical analysis has been performed and the information is added in the revised manuscript.

Comment: Can the authors explain why they chose BALB/c mice for their experiments? The study that the authors mention in their discussion (line 305) used C57BL/6 mice. Could that have potentially affected the Th1/Th2 responses knowing that the two mouse strains have a different Th bias? Perhaps discussion should be expanded to include these points.

Response: BALB/c mice have been extensively used in TB immunology/vaccine research. Some of the papers published in 2019 and available in PubMed are given below. 

1. Broset E, Saubi N, Guitart N, Aguilo N, Uranga S, Kilpeläinen A, Eto Y, Hanke T, Gonzalo-Asensio J, Martín C, Joseph-Munné J. MTBVAC-Based TB-HIV Vaccine Is Safe, Elicits HIV-T Cell Responses, and Protects against Mycobacterium tuberculosis in Mice. Mol Ther Methods Clin Dev. 2019 Feb 7;13:253-264.

2. Xiao TY, Liu HC, Li XQ, Huang MX, Li GL, Li N, Yan YH, Luo Q, Wang XZ, Li MC, Wan KL. Immunological Evaluation of a Novel Mycobacterium tuberculosis Antigen Rv0674. Biomed Environ Sci. 2019 Jun;32(6):427-437.

3. Nagpal PS, Kesarwani A, Sahu P, Upadhyay P. Aerosol immunization by alginate coated mycobacterium (BCG/MIP) particles provide enhanced immune response and protective efficacy than aerosol of plain mycobacterium against M.tb. H37Rv infection in mice.

BMC Infect Dis. 2019 Jul 1;19(1):568. 

4. Komine-Aizawa S, Jiang J, Mizuno S, Hayakawa S, Matsuo K, Boyd LF, Margulies DH, Honda M. MHC-restricted Ag85B-specific CD8+ T cells are enhanced by recombinant BCG prime and DNA boost immunization in mice. Eur J Immunol. 2019 Sep;49(9):1399-1414.

5. Moreno-Mendieta S, Barrera-Rosales A, Mata-Espinosa D, Barrios-Payán J, Sánchez S, Hernández-Pando R, Rodríguez-Sanoja R. Raw starch microparticles as BCG adjuvant: Their efficacy depends on the virulence of the infection strains.

Vaccine. 2019 Sep 10;37(38):5731-5737. 

6. Fitzpatrick M, Ho MM, Clark S, Dagg B, Khatri B, Lanni F, Williams A, Brennan M, Laddy D, Walker B. Comparison of pellicle and shake flask-grown BCG strains by quality control assays and protection studies. Tuberculosis (Edinb). 2019 Jan;114:47-53. 

7. Sawutdeechaikul P, Cia F, Bancroft G, Wanichwecharungruang S, Sittplangkoon C, Palaga T. Oxidized Carbon Nanosphere-Based Subunit Vaccine Delivery System Elicited Robust Th1 and Cytotoxic T Cell Responses. J Microbiol Biotechnol. 2019 Mar 28;29(3):489-499. 

8. Okay S, Çetin R, Karabulut F, Doğan C, Sürücüoğlu S, Kızıldoğan AK.

Immune responses elicited by the recombinant Erp, HspR, LppX, MmaA4, and OmpA proteins from Mycobacterium tuberculosis in mice. Acta Microbiol Immunol Hung. 2019 Jun 1;66(2):219-234.

9. Bull NC, Stylianou E, Kaveh DA, Pinpathomrat N, Pasricha J, Harrington-Kandt R, Garcia-Pelayo MC, Hogarth PJ, McShane H.

Enhanced protection conferred by mucosal BCG vaccination associates with presence of antigen-specific lung tissue-resident PD-1+ KLRG1- CD4+ T cells. Mucosal Immunol. 2019 Mar;12(2):555-564. 

Comment: Could that have potentially affected the Th1/Th2 responses knowing that the two mouse strains have a different Th bias? Perhaps discussion should be expanded to include these points.

Response: This part of the discussion has been removed in light of the comments by the Reviewer 1.

Minor comments:

Comment: The quality of figure 3 is really low. Would it be possible to replace with higher resolution pictures?

Response: Figure 3 has been replaced with higher resolution meeting the standard of PLOs ONE.

Comment: BCG – abbreviation should be explained

Response: BCG is abbreviation of Bacillus Calmette Guerin. It has been explained in the revised manuscript.

Comment: line 286 in discussion. What do the authors mean by safer? Subunit vaccines will have to be administered with or as a boost to BCG. How does that make them safer compared to BCG alone?

Response: The word “safer” has been deleted.

Comment: Line 361 Can the authors clarify which other stain do they compare the C57BL/6 mice to? And can they provide a reference to support this statement?

Response: This part of discussion has been deleted as suggested by Referee 1.

---

## [Decision Letter · Decision Letter 1]

2 Jan 2020

PONE-D-19-14947R1

The effect of adjuvants and delivery systems on Th1, Th2, Th17 and Treg cytokine responses in mice immunized with Mycobacterium tuberculosis-specific proteins

PLOS ONE

Dear Dr. Mustafa,

Thank you for submitting your manuscript to PLOS ONE. After careful consideration, we feel that it has merit but does not fully meet PLOS ONE’s publication criteria as it currently stands. Therefore, we invite you to submit a revised version of the manuscript that addresses the points raised during the review process.

Please attend to the one comment made by reviewer 1 regarding the statistical analyses and the comments  made by reviewer 2 regarding the use of immunogenicity data when there is no validated immune correlate of protection and the other technical points raised by this reviewer.

We would appreciate receiving your revised manuscript by Feb 16 2020 11:59PM. To enhance the reproducibility of your results, we recommend that if applicable you deposit your laboratory protocols in protocols.io, where a protocol can be assigned its own identifier (DOI) such that it can be cited independently in the future. For instructions see: http://journals.plos.org/plosone/s/submission-guidelines#loc-laboratory-protocols

We look forward to receiving your revised manuscript.

Kind regards,

Ann Rawkins, PhD

Academic Editor

PLOS ONE

Reviewers' comments:

Reviewer's Responses to Questions

**Comments to the Author**

1. If the authors have adequately addressed your comments raised in a previous round of review and you feel that this manuscript is now acceptable for publication, you may indicate that here to bypass the “Comments to the Author” section, enter your conflict of interest statement in the “Confidential to Editor” section, and submit your "Accept" recommendation.

Reviewer #1: All comments have been addressed

Reviewer #2: (No Response)

2. Is the manuscript technically sound, and do the data support the conclusions?

Reviewer #1: Yes

Reviewer #2: Partly

3. Has the statistical analysis been performed appropriately and rigorously? 

Reviewer #1: Yes

Reviewer #2: I Don't Know

4. Have the authors made all data underlying the findings in their manuscript fully available?

Reviewer #1: Yes

Reviewer #2: Yes

5. Is the manuscript presented in an intelligible fashion and written in standard English?

Reviewer #1: Yes

Reviewer #2: Yes

6. Review Comments to the Author

Reviewer #1: According to the comments, the manuscript was properly revised with improvement. I have one minor comment on statistics. Please clarify the objectives of comparision in all figures.

Reviewer #2: -The authors have addressed some comments but they still need to make it clearer in their discussion how measuring only immune responses rather than protection is a limitation due to lack of immune correlates in TB.

-Affinity columns will not remove endotoxin for the protein preparations. The authors should indicate at least the approximate levels of LPS. Big differences in endotoxin levels between proteins should be noted.

-Thanks to the authors for including antibody data. Can the authors clarify whether the serum was collected from the one experiment or whether sera from different experiments were used to create Figure 8.

-The comment about the use of BALB/c mice and potential Th bias has not been adequately addressed

-Figure 6: It is not entirely clear what the data presented are. Have the authors pooled all the data for the same protein from different experiments? If this is the case, how do they account for the huge differences in the non-stimulated controls between experiments? I would remove this figure as it is not scientifically accurate to pool responses from non-direct replicate experiments into one bar.

7. PLOS authors have the option to publish the peer review history of their article (what does this mean?). If published, this will include your full peer review and any attached files.

Reviewer #1: No

Reviewer #2: No

---

## [Author Response · Author response to Decision Letter 1]

9 Jan 2020

Response to reviewer’s comments

Reviewer #1: 

Comment: According to the comments, the manuscript was properly revised with improvement. I have one minor comment on statistics. Please clarify the objectives of comparison in all figures.

Response: We would like to thank the reviewer for stating that the manuscript was properly revised with improvement. 

The objective of comparison in all figures was to determine which immunization followed by stimulation of spleen cells in vitro resulted in secretion of significant concentrations of the individual cytokines. This information has been added in the materials and methods under data analysis.

Reviewer #2: -

Comment: The authors have addressed some comments but they still need to make it clearer in their discussion how measuring only immune responses rather than protection is a limitation due to lack of immune correlates in TB.

Response: We thank the referee for stating that the authors have addressed some comments. Furthermore, as per the suggestion of the reviewer, we have added in the end of discussion how measuring only immune responses rather than protection is a limitation due to lack of immune correlates in TB.

Comment: Affinity columns will not remove endotoxin for the protein preparations. The authors should indicate at least the approximate levels of LPS. Big differences in endotoxin levels between proteins should be noted.

Response: We had not measured the concentration of endotoxin in our recombinant protein preparations and hence it is not possible to indicate the approximate levels of LPS. However, we have used purified recombinant proteins only for immunizations using IFA and Alum as adjuvants. Other immunizations using recombinant mycobacteria (r M. vaccae and rM. smegmatis) and rDNA vaccine constructs will not have a problem of LPS contamination. The endotoxin contamination would have been a major concern in our study if we had used the recombinant proteins for in vitro stimulation of spleen cells for cytokine secretions. However, to avoid the non-specific stimulation of spleen cells due to the endotoxin contamination of the recombinant proteins, we have used chemically synthesized peptides covering the sequences of the individual proteins, as stated in the materials and methods. 

Comment: Thanks to the authors for including antibody data. Can the authors clarify whether the serum was collected from the one experiment or whether sera from different experiments were used to create Figure 8?

Response: The serum samples were collected from the one experiment to create Figure 8 (Figure 7 in the revised manuscript). 

Comment: The comment about the use of BALB/c mice and potential Th bias has not been adequately addressed.

Response: We agree with the comment of the reviewer that the use of BALB/c mice and the potential Th bias was not adequately addressed. The reason being that the experimental studies have shown divergent results on the Th responses in response to mycobacterial antigens in mice of different genetic backgrounds as given below.

A. Examples of studies showing that C57BL/6 mice have Th1 profile and BALB/c mice have Th2.

1. Huygen K, Abramowicz D, Vandenbussche P, Jacobs F, De Bruyn J, Kentos A, Drowart A, Van Vooren JP, Goldman M. Spleen cell cytokine secretion in Mycobacterium bovis BCG-infected mice. Infect Immun. 1992 Jul;60(7):2880-6.

2. Wakeham J, Wang J, Xing Z. Genetically determined disparate innate and adaptive cell-mediated immune responses to pulmonary Mycobacterium bovis BCG infection in C57BL/6 and BALB/c mice.

Infect Immun. 2000 Dec;68(12):6946-53.

3. Paula MO, Fonseca DM, Wowk PF, Gembre AF, Fedatto PF, Sérgio CA, Silva CL, Bonato VL.Host genetic background affects regulatory T-cell activity that influences the magnitude of cellular immune response against Mycobacterium tuberculosis. Immunol Cell Biol. 2011 May;89(4):526-34.

4. Sérgio CA, Bertolini TB, Gembre AF, Prado RQ, Bonato VL. CD11c(+) CD103(+) cells of Mycobacterium tuberculosis-infected C57BL/6 but not of BALB/c mice induce a high frequency of interferon-γ- or interleukin-17-producing CD4(+) cells.

Immunology. 2015 Apr;144(4):574-86. 

B. Examples of studies showing that BALB/c mice have a stronger Th1 and Th17 responses than C57BL/6 mice or both mouse strains have similar Th profiles.

1. Garcia-Pelayo MC, Bachy VS, Kaveh DA, Hogarth PJ. BALB/c mice display more enhanced BCG vaccine induced Th1 and Th17 response than C57BL/6 mice but have equivalent protection. Tuberculosis (Edinb). 2015;95(1):48-53.

2. Leung-Theung-Long S, Gouanvic M, Coupet CA, Ray A, Tupin E, Silvestre N, Marchand JB, Schmitt D, Hoffmann C, Klein M, Seegren P, Huaman MC, Cristillo AD, Inchauspé G. A Novel MVA-Based Multiphasic Vaccine for Prevention or Treatment of tuberculosis induces broad and multifunctional cell-mediated immunity in mice and primates. PLoS One. 2015;10(11):e0143552.

3. Yadav B, Malonia SK, Majumdar SS, Gupta P, Wadhwa N, Badhwar A, Gupta UD, Katoch VM, Chattopadhyay S. Constitutive expression of SMAR1 confers susceptibility to Mycobacterium tuberculosis infection in a transgenic mouse model. Indian J Med Res. 2015 Dec;142(6):732-41.

4. Walton CB, Inos AB, Andres OA, Jube S, de Couet HG, Douglas JT, Patek PQ, Borthakur D. Immunization with hybrid recombinant Mycobacterium tuberculosis H37Rv proteins increases the TH1 cytokine response in mice following a pulmonary instillation of irradiated mycobacteria. Vaccine. 2008;26(34):4396-402.

5. Roupie V, Romano M, Zhang L, Korf H, Lin MY, Franken KL, Ottenhoff TH, Klein MR, Huygen K. Immunogenicity of eight dormancy regulon-encoded proteins of Mycobacterium tuberculosis in DNA-vaccinated and tuberculosis-infected mice. Infect Immun. 2007;75(2):941-9.

Hence, selection of BALB/c mice based on a specific Th profile does not seem to be justified and, therefore, we did not address this issue. 

Comment: Figure 6: It is not entirely clear what the data presented are. Have the authors pooled all the data for the same protein from different experiments? If this is the case, how do they account for the huge differences in the non-stimulated controls between experiments? I would remove this figure as it is not scientifically accurate to pool responses from non-direct replicate experiments into one bar.

Response: Figure 6: Yes, all the data were pooled for the same protein from different experiments. We agree with the reviewer and Figure 6 has been removed in the revised manuscript.

---

## [Decision Letter · Decision Letter 2]

15 Jan 2020

The effect of adjuvants and delivery systems on Th1, Th2, Th17 and Treg cytokine responses in mice immunized with Mycobacterium tuberculosis-specific proteins

PONE-D-19-14947R2

Dear Dr. Mustafa,

We are pleased to inform you that your manuscript has been judged scientifically suitable for publication and will be formally accepted for publication once it complies with all outstanding technical requirements.

With kind regards,

Ann Rawkins, PhD

Academic Editor

PLOS ONE

Additional Editor Comments (optional):

Reviewers' comments:

Reviewer's Responses to Questions

**Comments to the Author**

1. If the authors have adequately addressed your comments raised in a previous round of review and you feel that this manuscript is now acceptable for publication, you may indicate that here to bypass the “Comments to the Author” section, enter your conflict of interest statement in the “Confidential to Editor” section, and submit your "Accept" recommendation.

Reviewer #2: All comments have been addressed

2. Is the manuscript technically sound, and do the data support the conclusions?

Reviewer #2: Yes

3. Has the statistical analysis been performed appropriately and rigorously? 

Reviewer #2: I Don't Know

4. Have the authors made all data underlying the findings in their manuscript fully available?

Reviewer #2: Yes

5. Is the manuscript presented in an intelligible fashion and written in standard English?

Reviewer #2: Yes

6. Review Comments to the Author

Reviewer #2: (No Response)

7. PLOS authors have the option to publish the peer review history of their article (what does this mean?). If published, this will include your full peer review and any attached files.

Reviewer #2: No

---

## [Editor Report · Acceptance letter]

21 Jan 2020

PONE-D-19-14947R2 

The effect of adjuvants and delivery systems on Th1, Th2, Th17 and Treg cytokine responses in mice immunized with *Mycobacterium tuberculosis*-specific proteins 

Dear Dr. Mustafa:

I am pleased to inform you that your manuscript has been deemed suitable for publication in PLOS ONE. Congratulations! Your manuscript is now with our production department. 

With kind regards,

on behalf of

Dr. Ann Rawkins 

Academic Editor

PLOS ONE